# Ferritin, blood urea nitrogen, and high chest CT score determines ICU admission in COVID-19 positive UAE patients: A single center retrospective study

**Riyad Bendardaf[1,2☉]\*, Poorna Manasa Bhamidimarri[2,3☉], Zainab Al-Abadla[4], Dima Zein[5], Noura Alkhayal[6], Ramy Refaat Georgy[7], Feda Al Ali[5], Alaa Elkhider[5], Sadeq Qadri[5], Rifat Hamoudi[2,8]\*, Salah Abusnana[2,4]**

**1** Department of Oncology, University Hospital Sharjah, Sharjah, United Arab Emirates, **2** Clinical Sciences Department, College of Medicine, University of Sharjah, Sharjah, United Arab Emirates, **3** Sharjah Research Academy, Sharjah, United Arab Emirates, **4** Department of Diabetes and Endocrinology, University Hospital Sharjah, Sharjah, United Arab Emirates, **5** Department of Internal Medicine, University Hospital Sharjah, Sharjah, United Arab Emirates, **6** Department of Medical Laboratory, University Hospital Sharjah, Sharjah, United Arab Emirates, **7** Department of Medical Diagnostic Imaging, University Hospital Sharjah, Sharjah, United Arab Emirates, **8** Division of Surgery and Interventional Science, University College London, United Kingdom

☉ These authors contributed equally to this work.
\* Riyad.Bendardf@uhs.ae (RB); rhamoudi@sharjah.ac.ae (RH)

**Data Availability Statement:** All relevant data are within the paper and its Supporting information files.

## Abstract

Coronavirus Disease (COVID-19) was declared a pandemic by WHO in March 2020. Since then, additional novel coronavirus variants have emerged challenging the current healthcare system worldwide. There is an increased need for hospital care, especially intensive care unit (ICU), for the patients severely affected by the disease. Most of the studies analyzed COVID-19 infected patients in the hospitals and established the positive correlation between clinical parameters such as high levels of D-dimer, C-reactive protein, and ferritin to the severity of infection. However, little is known about the course of the ICU admission. The retrospective study carried out at University Hospital Sharjah, UAE presented here reports an integrated analysis of the biochemical and radiological factors among the newly admitted COVID-19 patients to decide on their ICU admission. The descriptive statistical analysis revealed that patients with clinical presentations such as acute respiratory distress syndrome (ARDS) (p<0.0001) at the time of admission needed intensive care. The ROC plot indicated that radiological factors including high chest CT scores (>CO-RADS 4) in combination with biochemical parameters such as higher levels of blood urea nitrogen (>6.7 mg/dL;66% sensitivity and 75.8% specificity) and ferritin (>290 µg/mL, 71.4% sensitivity and 77.8% specificity) may predict ICU admission with 94.2% accuracy among COVID-19 patients. Collectively, these findings would benefit the hospitals to predict the ICU admission amongst COVID-19 infected patients.

**Funding:** The current study was funded by University of Sharjah (Grant codes. CoV19-0308, CoV19-03010; Sharjah, UAE); Sharjah Research Academy (Grant No: MED001); University of Sharjah (Grant No: 1901090254). The funders had no role in study design, data collection and analysis, decision to publish, or preparation of the manuscript.

**Competing interests:** The authors have declared that no competing interests exist.

## 1. Introduction

Coronavirus disease (COVID-19), caused by novel coronavirus (CoV) was first identified at Wuhan, China in December 2019 [1]. This new pathogen causes respiratory complications and proved to be contagious with its fast spread to more than 219 countries worldwide. World Health Organization (WHO) declared COVID-19 as a global pandemic in March 2020 [2]. Since its outbreak, CoV has infected more than 124 million people and caused nearly three million deaths worldwide as of March 2021 (https://www.worldometers.info/coronavirus/). Although the death rate is under control, the emergence of the new variants of the virus and the anticipated second and third waves of infection has raised concerns in the clinical community to prepare the hospital facilities [3–5].

In United Arab Emirates (UAE), the first COVID-19 case was reported in January 2020 [6] and the current number of infected cases reached to 440,000 with approximately 1500 deaths reported till date. The containment of the spread was effective in this region, however, as a part of the second wave of COVID-19 infection there is an increase in the number of deaths reported in the last five months. In the current scenario where, new variants spread across countries worldwide, there is a need to identify key factors to effectively and expeditiously control the spread of COVID-19 infection. The current challenges to combat the pandemic includes: identification of infected patients in the initial stages, to predict the level of severity of COVID-19 infection, and provide the correct clinical management guidelines to the hospitals [7].

Studies performed initially in China and in the early affected countries have described the important clinical and biochemical characteristics in COVID-19 infected patients [8–11]. These studies analyzed the demographic and clinical data of the patients and informed the increased risk of death among aged males. Comorbidities such as diabetes and hypertension were predicted as risk factors for severe infection [12, 13]. Biochemical factors such as higher levels of d-dimer, c-reactive protein, and ferritin were reported to be indicators of COVID-19 infection [14, 15]. Pulmonary and extrapulmonary inflammation were associated with severe COVID-19 infections which led to multi-organ dysfunction and eventually death [16]. Symptoms such as respiratory failure, acute cardiac injury, and ARDS were considered as critical factors to predict the risk of death in severe COVID-19 infected patients [17, 18].

In UAE, studies were conducted that could establish a similar profile of biomarkers as reported worldwide [19]. Few case-control studies were performed concentrating on a particular clinical condition like diabetes, cardiovascular disease, and neurological disorders [6, 20]. Very limited studies were performed on the ICU-admitted COVID-19 patients. The studies available so far only analyzed a focused cohort based on a pre-existing condition or a biochemical parameter [12, 21]. To the best of our knowledge, this is the first study to investigate the association of radiological findings with biochemical characteristics that demarcate ICU admitted COVID-19 infected patients.

The present report is a retrospective study of the COVID-19 infected patients admitted during the early period of the pandemic from February to August 2020 in the University Hospital Sharjah, UAE.

## 2. Methods

### 2.1. Patient recruitment and ethics

This study is a retrospective, observational analysis to define the biochemical, radiological, and clinical characteristics of the COVID-19 positive patients admitted at University Hospital Sharjah, Sharjah, UAE between February to August 2020. In total 106 patients were considered

for the study. Of which 40 patients recovered without the need of ICU admission and 66 patients were admitted to ICU.

Ethical approval was obtained for the study from University Hospital Sharjah with reference number UHS-HERC- 037–21062020. Patient information was anonymized and the data required for the study was collected. In accordance with institutional ethical considerations, informed consent from the patients was not required for this study.

### 2.2. Data collection

The demographic data (gender, age, Body Mass Index) along with routine blood profile tests were collected from the medical records. The clinical features recorded include comorbidities as hypertension, diabetes, cardiovascular, cerebrovascular, chronic kidney disease and others; symptoms like fever, cough, fatigue, myalgia, dyspnea, chest tightness, Pharyngalgia, diarrhea, nausea, vomiting, abdominal pain, headache and disordered consciousness; clinical complications include acute respiratory distress syndrome (ARDS), Type1 respiratory failure, sepsis, acidosis, alkalosis, acute kidney injury, hyperkaliemia, SARS, CRS and acute liver injury; laboratory profiles like Hemoglobin, Platelet count, WBC count, Neutrophil, Lymphocyte Count, Monocyte Count, alanine aminotransferase, albumin, alkaline phosphatase, aspartate aminotransferase, total bilirubin, Potassium, Sodium, blood urea nitrogen, creatinine, lactate dehydrogenase, Hypersensitive cardiac troponin, Prothrombin time, International Normalized Ratio (INR), activated partial thromboplastin time, D-dimer, procalcitonin, CRP, ferritin, pH, pCO2, pO2, bicarbonate and base excess of blood.

### 2.3. Computed Tomography (CT) and CO-RADS (COVID-19 Reporting and Data System)

The images were obtained from High-Resolution CT (HRCT) imaging of the chest and scored by a radiologist for COVID-19 Reporting and Data System (CO-RADS) classification at University Hospital Sharjah, Sharjah, UAE. The representative images were provided in Fig 1 and Table 1 informs the scoring based on CT image. Higher the score the more is the positive outcome of COVID-19 disease, which also correlates with the severity.

### 2.4. Statistical analysis

Descriptive statistics were carried out to characterize the cohort. The differential analysis between the non-ICU and ICU admitted cases was performed using the Chi-square test for categorical variables and Student's t-test for continuous variables. P-value < 0.05 was considered as significant. All statistical analysis was carried out using SPSS software (version 23). A receiver operating characteristic curve (ROC) was plotted to test the association between biochemical and radiological findings from the study that could predict ICU admission. Sensitivity and specificity for the cut-off values were analyzed from the plot and the accuracy was measured using the formula [(True Negative +True Positive)/ (False Positive +True Positive +True Negative +False Negative)] and presented as a percentage.

## 3. Results

### 3.1. Demographic data and grouping into ICU and non-ICU

The demographic and basic clinical variables for all the patients were detailed in Table 2. Among the 106 COVID-19 positive patients considered for the study, 40 cases recovered without the need to be admitted to ICU. Out of 66 cases admitted to ICU, around 31 (47%) cases died and only 35 (53%) survived even upon intensive care management. Among the 40

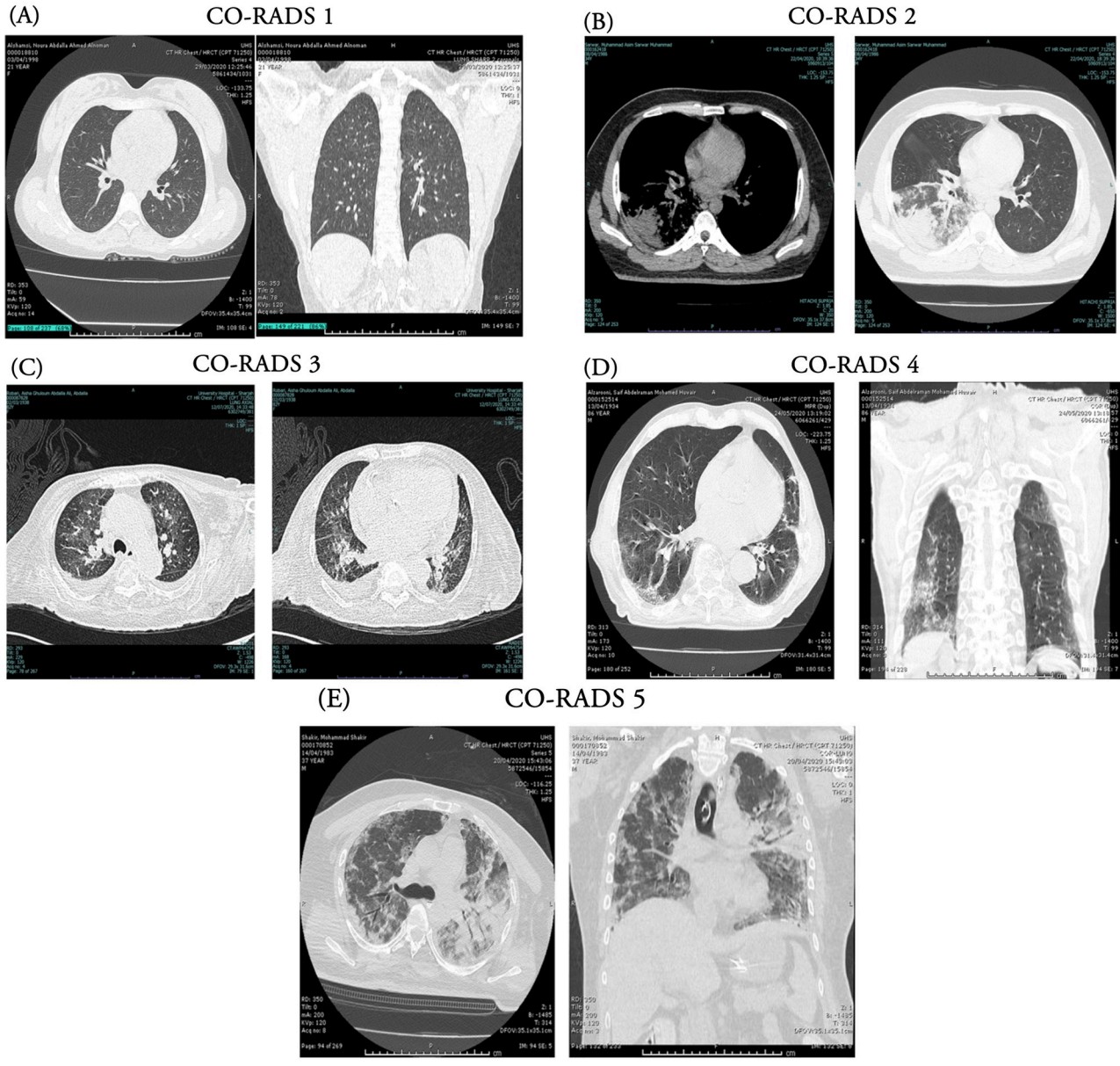

**Fig 1. Computed Tomography (CT)-scan images showing the scoring for CO-RADS based on the opacity observed in lungs.** (A) CO-RADS 1 (B) CO-RADS 2 (C) CO-RADS 3 (D) CO-RADS 4 (E) CO-RADS 5.

**Table 1. COVID-19 Reporting and Data System (CO-RADS) based demarcation among the COVID-19 positive patients.**

| CO-RADS score | Classification | non-ICU admitted N (%) | ICU admitted N (%) |
|---|---|---|---|
| CO-RADS 1 | Low | 5 (12.5) | 0 |
| CO-RADS 2 | Intermediate | 5 (12.5) | 2 (3) |
| CO-RADS 3 | | | |
| CO-RADS 4 | High | 22 (55) | 61 (92.4) |
| CO-RADS 5 | | | |
| | No data | 8 (20) | 3 (4.5) |

**Table 2. Demographic characteristics of the COVID-19 positive patients demarcated into non-ICU and ICU admitted.**

| Variable | non-ICU admitted Total 40 | ICU admitted Total 66 |
|---|---|---|
| Gender; N (%) | | |
| Male | 15 (37.5) | 11 (65) |
| Female | 25 (62.5) | 6 (35) |
| Age | | |
| Average; mean ± SD | 55±20 | 63±15 |
| Range; years | 21–94 | 34–84 |
| Discharge Outcome; N (%) | | |
| Alive | 40 (100%) | 35 (53) |
| Deceased | 0 | 31 (47) |
| BMI; mean ± SD | 27±5 | 27.6±5.7 |
| Temperature; mean ± SD | 37.4±0.8 | 37.4±0.8 |
| Heart Rate; mean ± SD | 89.8±18 | 91±16.5 |
| Breaths per minute; mean ± SD | 19.1±2.6 | 20.4±2.9 |
| Oxygen Saturation; mean ± SD | 95.4±3 | 89.6±6.9 |

patients recovered, 15 (37.5%) are male and 25 (62.5%) are female whereas 56% of ICU admitted cases are male. The average age (mean ± SD) among the non-ICU admitted is 55±20 years and in ICU admitted group is 63±15 years. Oxygen saturation dropped slightly in the case of ICU admitted patients. Only 11 (27.5%) among non-ICU and 6 (9%) among ICU admitted known to have a previous contact history with COVID patients.

## 3.2. Clinical characteristics and laboratory findings among the COVID-19 patients at the time of admission

**3.2.1. Comorbidities.** Data suggested that 12 (30%) in the non-ICU group and 12 (18.8%) in the ICU group reported no comorbid condition. Almost 50% of non-ICU cases and 67% of the ICU admitted patients reported a history of hypertension. Comorbidities like cardiovascular and cerebrovascular complications, Hyperlipidemia, and Dyslipidemia were reported by less than 30% of cases among both the groups. Of the severe patients admitted to ICU, 53% and 27% cases reported Diabetes and Chronic kidney disease respectively (S1 Table). Comorbidities that can demarcate non-ICU and ICU admitted groups were shown in Fig 2.

**3.2.2. Clinical symptoms.** All the clinical symptoms recorded for the COVID-19 patients admitted to the hospital were listed in Table 3. The most common symptoms in both the groups include fever and cough as reported by more than 60% patients in each group. Other common symptoms presented by less than 25% cases in both the groups include fatigue, myalgia, Pharyngalgia, abdominal pain, vomiting, diarrhea, and headache. Important symptoms like dyspnea, chest tightness, acute respiratory distress syndrome (ARDS), type 1 respiratory failure, alkalosis, and acute kidney injury were mostly reported in severe patients admitted to ICU.

**3.2.3. Biochemical characteristics.** Biochemical findings were sorted into subgroups like Blood Profile, Renal profile, Liver functionality, Inflammatory profile, Coagulation profile, Acid/base analysis, and others. All the values were presented as mean ± SD in Table 4. Within the blood profile, there was no major difference in the non-severe and severe cases except for a decrease in hemoglobin levels and neutrophilia in ICU admitted cases. In the renal profile, blood urea nitrogen and creatinine levels were very high in severe cases.

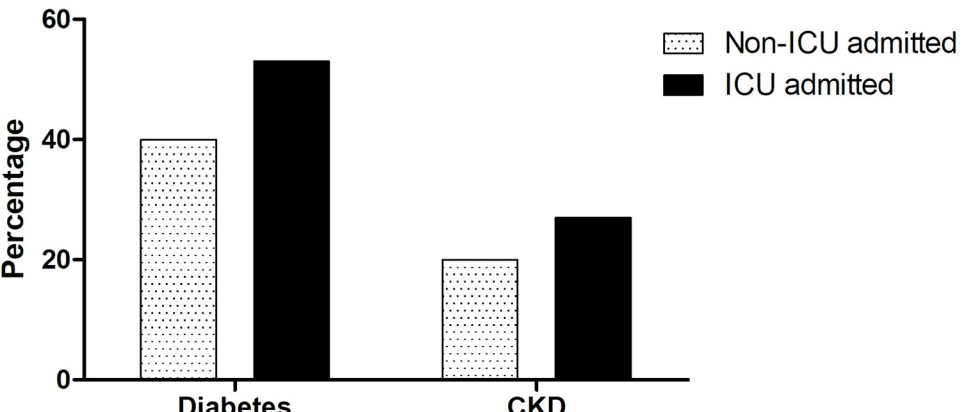

**Fig 2. Bar graph representing the significant comorbidities among the COVID-19 positive patients admitted.**

The liver functionality test report distinctly differentiates the ICU admitted group with the albumin, bilirubin and alkaline phosphatase levels severely altered. The inflammatory profile proved to be the best indicator of the severity of the cases with the levels of c-reactive protein, procalcitonin, ferritin and Lactate dehydrogenase (LDH) shooting up in ICU admitted patients. Activated partial thromboplastin time delayed and the D-dimer levels were high in ICU admitted patients within the coagulation profile. The bicarbonate value in non-ICU cases was 18.9±6.7 and in the ICU admitted cases was 25.2±4.4 indicating alkalosis among ICU admitted patients. The important biochemical factors that are highly significant in distinguishing the ICU admitted cases from the non-ICU admitted patients were shown in Fig 3.

**3.2.4. Radiological findings.** Computed tomography (CT) of the chest was used to develop a COVID-19 Reporting and Data System (CO-RADS). CO-RADS with higher values 4 and 5 were prominently seen in ICU admitted cases (92.4%) than non-ICU admitted cases (55%) (Fig 4 and Table 1). Only 2 patients were diagnosed with the intermediate score in ICU admitted group.

## 3.3. Clinical management and treatment in ICU

In general, the guidelines provided by the Ministry of Health (MOH), UAE were followed to manage the COVID-19 patients. As per the guidelines, mild and asymptomatic cases were asked to isolate at home, and moderate to severe cases were admitted to the hospital in

**Table 3. Symptoms upon admission among the COVID-19 positive patients demarcated into non-ICU and ICU admitted.**

| Symptoms | non-ICU admitted N (%) | ICU admitted N (%) | p value |
|---|---|---|---|
| Fever | 29 (72.5) | 55 (83) | 0.182 |
| Cough | 25 (62.5) | 50 (75) | 0.146 |
| Fatigue | 6 (15) | 40 (60.6) | **<0.0001** |
| Myalgia | 9 (22.5) | 37 (56) | 0.001 |
| Dyspnea | 11 (27.5) | 51 (77) | **<0.0001** |
| Chest tightness | 1 (2.5) | 20 (30) | **<0.0001** |
| Disordered Consciousness | 1 (2.5) | 13 (19.6) | **0.015** |
| ARDS | 10 (25) | 33 (50) | **0.014** |
| Acidosis | 1 (2.5) | 10 (15.1) | **0.049** |
| Alkalosis | 0 | 20 (30) | **<0.0001** |

**Table 4. Biochemical and laboratory findings of the COVID-19 positive patients demarcated into non-ICU and ICU admitted.**

| Laboratory parameter | non-ICU admitted | ICU admitted | p value |
|---|---|---|---|
| **Blood Profile; mean±SD** | | | |
| Hemoglobin (g/dL) | 12.3±1.8 | 11.91±2.13 | 0.378 |
| Platelet Count (x10$^9$/L) | 217.8±76.7 | 231.7±91.9 | 0.438 |
| WBC Count (x10$^9$/L) | 6.2±2.2 | 8.1±4.2 | **0.029** |
| Neutrophil (x10$^9$/L) | 4±1.8 | 6.4±4.1 | **0.001** |
| Lymphocyte Count (x10$^9$/L) | 1.5±0.9 | 1.75±2.4 | 0.305 |
| Monocyte Count (x10$^9$/L | 0.5±0.2 | 0.86±1.9 | 0.581 |
| **Renal profile; mean±SD** | | | |
| Potassium (mmol/L) | 4±0.52 | 4.1±0.74 | 0.895 |
| Sodium (mmol/L) | 134.7±3.4 | 135.2±6.3 | 0.976 |
| Blood Urea Nitrogen (mg/dL) | 6.6±8 | 10.7±7.1 | **<0.0001** |
| Creatinine (umol/L) | 119.7±175.8 | 177.4±220.7 | **0.049** |
| **Liver Functionality; mean±SD** | | | |
| Alanine Aminotransferase (U/L) | 33.3±19.6 | 143.45±471 | 0.128 |
| Albumin (g/L) | 31.2±5.35 | 33±6 | 0.323 |
| Alkaline Phosphatase (IU/L) | 68.2±23.1 | 100±54.1 | **0.001** |
| Aspartate Aminotransferase (U/L) | 34±32.15 | 367.7±2374 | **<0.0001** |
| Total Bilirubin (umol/L) | 9±7.4 | 11.35±7.9 | **0.006** |
| **Inflammatory Profile; mean±SD** | | | |
| C-Reactive Protein (nmol/L) | 73.1±102.6 | 122.12±99.5 | **<0.0001** |
| Procalcitonin (ug/L) | 0.68±3.2 | 10.45±59.7 | **<0.0001** |
| Ferritin (ug/mL) | 288.8±408 | 905.02±1146 | **<0.0001** |
| Lactate Dehydrogenase (U/L) | 236.8±79.7 | 412.23±303.6 | **<0.0001** |
| **Coagulation Profile; mean±SD** | | | |
| Prothrombin Time (s) | 14.5±1.5 | 15.1±2.8 | 0.274 |
| International Normalized Ratio (INR) | 1.1±0.15 | 1.15±0.3 | 0.281 |
| Activated Partial Thromboplastin (s) Time | 35.8±4.6 | 41.75±11.2 | **0.011** |
| D-dimer (ug/mL) | 1.1±1 | 3.12±5.4 | **0.006** |
| **Acid/base analysis; mean±SD** | | | |
| pH | 7.3±0.14 | 7.4±0.012 | 0.164 |
| pCO2 (mmHg) | 29.85±11.35 | 41.8±10 | 0.023 |
| pO2 (mmHg) | 81.12±37.3 | 61.8±30.25 | 0.171 |
| Bicarbonate (mmol/L) | 18.9±6.7 | 25.2±4.4 | **0.020** |
| Base Excess of Blood (mEq/L) | -8.6±11.3 | 7.71±50.33 | **0.002** |
| **Other; mean±SD** | | | |
| High-sensitivity Cardiac Troponin | 1.9±7.8 | 62.9±280.2 | **<0.0001** |

isolation rooms with contact and droplets precautions. Biochemical and physiological investigations were performed as detailed in laboratory findings earlier. Most of the cases (irrespective of severity) were provided with either of the antibiotics like Ceftriaxone, Amoxiclav, Amoxicillin, Azithromycin, Tazocin, Zinnat, Ceftazidime, Meropenem, Piperacillin Sodium, Tazobactam Sodium, Levofloxacin, Clindamycin, Vancomycin, Piperacillin, Gentamicin, Tobramycin, and Azomycin (>80% in both groups) and hydroxychloroquine (>50% in both groups). Antiviral therapy (Kaletra or Favipiravir or both) was provided in both the cases but predominantly given to ICU admitted cases due to the severity. Remidisivir has been authorized by FDA for emergency use during the pandemic along with Favipiravir. MOH recommends the use of Favipiravir treatment to be started at 1600 mg BID loading doses followed by

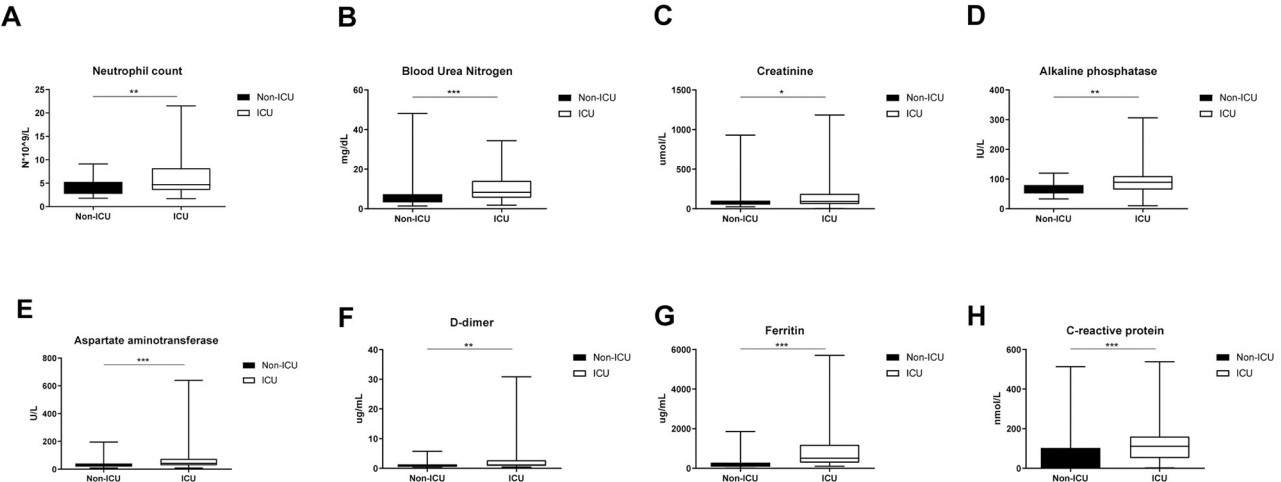

**Fig 3. Boxplots for highly significant biochemical characteristics demarcating non-ICU and ICU admitted COVID-19 positive patients.** Boxplot for (C) Creatinine showed a slight increase in their levels among ICU admitted group where '*' denotes p<0.05. Boxplot for (A) Neutrophil count, (D) Alkaline phosphatase, and (F) D-dimer showed higher amounts in ICU admitted group for respective values where '**' denotes p<0.005. Boxplot for (B) Blood Urea Nitrogen, (E) Aspartate aminotransferase and (G) Ferritin and (H) C-reactive protein showed elevated levels in ICU admitted group where '***' denotes p<0.0001.

600 mg BID for 7 to 10 days. Supplements have been added in the form of vitamin C, D and zinc. Almost 22% of non-ICU and 60% of ICU admitted cases were given Glucocorticoid Therapy with drugs like Hydrocortisone, Dexamethasone, Methylprednisolone and Predniso-lone. Corticosteroids has a major role in COVID management and it showed benefits in COVID patients when initiated early in the course of illness. The details on the clinical management were listed in S2 Table.

According to MOH instructions, vital signs, and oxygen saturation are to be monitored with a pulse oximeter and if oxygen saturation drops below 94%, oxygen therapy must be initiated. ICU admitted patients received Oxygen treatment, high flow nasal cannula, and mechanical ventilation. These treatments were prerequisites for clinical management of ICU admitted.

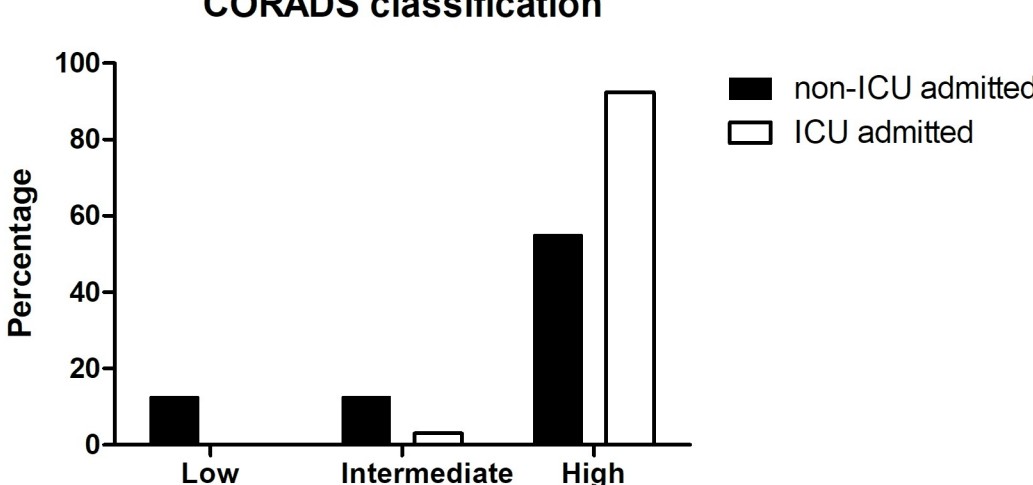

**Fig 4. Bar graph showing the CO-RADS distribution across the COVID-19 positive patients demarcated into non-ICU and ICU admitted.**

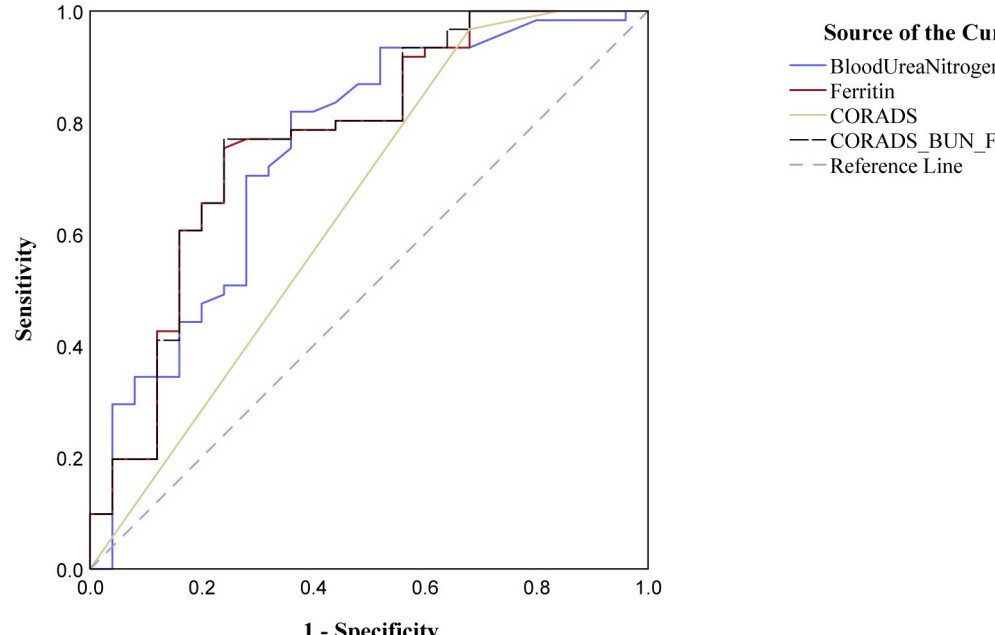

**Fig 5. Receiver operative characteristic curve plotted for CO-RADS, blood urea nitrogen and ferritin across the COVID-19 positive patients.** ICU admission is considered as positive actual state. The line with combination of CO-RADS, Blood urea nitrogen, and Ferritin showed the best fit among all the parameters tested.

### 3.4. Significant biochemical parameters that can predict the ICU admission

In order to identify the key indicators to predict the ICU admission requirement from the data available in the study, a receiver operating characteristic curve was plotted for CO-RADS score and the biochemical factors were significantly different among ICU and non-ICU admitted groups. It is evident earlier that the ICU admitted patients showed predominantly higher CO-RADS scores (Fig 4). The ROC plot presented here (Fig 5) revealed that the highly significant biochemical factors, determined before in this study (Table 3), blood urea nitrogen (AUC = 0.751) and ferritin (AUC = 0.770) independently differentiated ICU admission with considerable sensitivity and specificity (Table 5). However, the combination of CO-RADS score with the biochemical parameters proved to be the best fit in distinguishing ICU admitted patients with AUC = 0.772 (Table 5).

The sensitivity and specificity calculations based on the ROC plot for the variables independently showed CO-RADS score >4 gave the least specificity (37.5%) and a cut off value for blood urea nitrogen >6.7 mg/dL gave 66% sensitivity, 95% CI = 53.5–77.4%, and 75.8% specificity, 95% CI = 57.7–88.9%. Ferritin alone gave a good separation at cut off value >290 ug/mL

**Table 5. COVID-19 Reporting and Data System (CO-RADS) based ROC plot output for highly significant biochemical properties.**

| Variable | Area | Std error | Asymptomatic Sig | Asymptomatic 95% CI | |
|---|---|---|---|---|---|
| | | | | Lower bound | Upper bound |
| Blood Urea Nitrogen | 0.751 | 0.062 | 0.000 | 0.630 | 0.872 |
| Ferritin | 0.770 | 0.060 | 0.000 | 0.653 | 0.887 |
| CO-RADS | 0.646 | 0.072 | 0.034 | 0.506 | 0.787 |
| CO-RADS+BUN+Ferritin | 0.772 | 0.060 | 0.000 | 0.655 | 0.889 |

**Table 6. Sensitivity, specificity and accuracy for the three factors derived from the ROC plot.**

| Variable | Cutoff | Sensitivity (95% CI) | Specificity (95% CI) | Accuracy |
|---|---|---|---|---|
| Blood Urea Nitrogen (A) | >6.7 | 66% (53.3 to 77.4%) | 75.8% (57.7–88.9%) | 71.4% |
| Ferritin (B) | >290 | 71.4% (58.6 to 82.1%) | 77.8% (60.8–90%) | 75.7% |
| CO-RADS (C) | >4 | 93.6% (84.5 to 98.2%) | 37.5% (21.1 to 56.3%) | 74.7% |
| A+B+C | - | - | - | 94.2% |

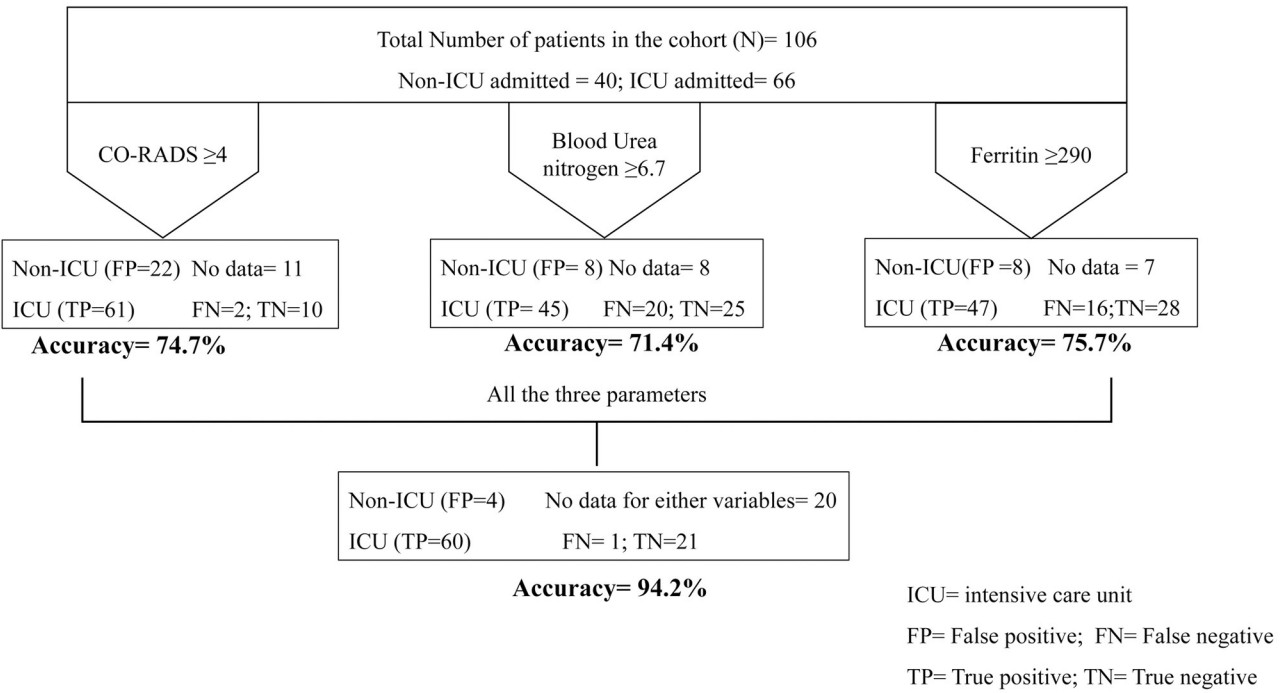

**Fig 6. Schematic for the accuracy of the parameters to demarcate ICU admission among COVID-19 patients.**

with 71.4% sensitivity, 95% CI = 58.6–82.1% and 77.8% specificity, 95% CI = 60.8–90% (Table 6). However, a cumulative analysis for all the three parameters showed a better area under curve 0.772 (Fig 5 and Table 5) avoiding any false positive detection with 94.2% accuracy (Fig 6 and Table 6). The false positives reduced with addition of CORADS to assess the ICU admission along with BUN and ferritin.

## 4. Discussion

The present observational study informed interesting correlations between the radiological and biochemical factors that distinguish the ICU admitted among the COVID-19 patients admitted to University Hospital Sharjah, UAE from February to August 2020. The data from the single center cohort of 106 patients considered for the study is in agreement with similar studies performed earlier at the time of the first wave of COVID-19 infection [15, 16, 18].

Almost 62% of the COVID-19 positive patients in the current cohort were admitted to ICU of which 47% patients died signifying a high death rate among ICU admitted patients. So as to provide early care and increase the chance of survival, the factors that demarcate the non-ICU admitted COVID-19 patients from ICU admitted cases were determined.

Most of the ICU admitted patients were Male (56%) with an average age 63±15 years. Similar observations were made in the studies performed in the region and worldwide [6, 19, 22–25]. A gender-based differential mortality risk for COVID-19 infection was attributed to the possible protective effect of estrogen among females as it is known to involve in immunomodulatory pathways [26]. COVID-19 infected patients reported with pre-existing comorbidities such as diabetes and chronic kidney disease (CKD) were at high risk to be admitted to ICU. The current study is one among the very few reports that indicated CKD is an important risk factor for severe COVID-19 infection [27].

Fever (77%) and cough (63%) are the most common symptoms reported by COVID-19 infected patients in this study. The key clinical symptoms that significantly differentiated ICU admitted patients from the non-ICU admitted were Dyspnea (p<0.0001) and ARDS (p = 0.014), similar to the observations made earlier from other retrospective studies [8, 19]. The present study also reports alkalosis and acute kidney injury as significant clinical symptoms among ICU admitted patients. These findings re-emphasize that CKD is an important risk factor as discussed earlier in the study.

The biochemical laboratory findings could demarcate the severely infected COVID-19 patients from mild, non-ICU admitted patients. Neutrophilia was observed in ICU admitted patients as expected from the previous analysis [28]. The role of neutrophils in acute lung injury was known and more specifically neutrophil activation among COVID-19 infected patients has been reported implicating the current finding [29]. Blood urea nitrogen and creatinine, the two important indicators of kidney function were highly altered in ICU admitted patients further validating the pre-existing CKD as a high-risk factor for severe COVID-19 infections.

Though the ICU admitted patients had no pre-existing liver complications, interestingly, the liver functionality profile was seen extremely altered among them. Albumin levels dropped while the values for liver enzymes alkaline phosphatase, aspartate aminotransferase, and total bilirubin increased from the normal. A recent review associated the COVID-19 infection with liver disease indicating that severe lung infection may lead to dysregulation of liver function through systemic inflammation [30]. Many retrospective studies reported the increase in the levels of liver enzymes in critically ill COVID-19 infected patients [31]. Consequently, the severely infected COVID-19 patients are at risk for developing liver dysfunction disorders, a possible reason for the higher number of deaths among ICU admitted patients.

The key inflammatory factors c-reactive protein and ferritin, reported in all the COVID-19 based studies [11, 14, 21] as indicators for severe infection, exhibited increased levels in ICU admitted patients. Ferritin levels were highly significant in demarcating ICU admitted patients from mild cases. In fact, a recent single center study identified ferritin as an independent factor predicting the mortality rate in COVID-19 infected patients [32]. Elevated levels of D-dimer and increased activated thromboplastin time (coagulative profile) among severely infected COVID-19 patients was detected, a key physiological effect of coronavirus infection. The increased bicarbonate levels and reduced base excess of blood were noticed in severe cases which can be associated with respiratory failure symptoms. These findings correlate with general observations related to lung infections [15] and so it was proposed that patients admitted with COVID-19 infection exhibiting elevated levels of ferritin, d-dimer, and c-reactive protein and low oxygen saturation values are managed with ICU protocol by WHO [33].

COVID-19 Reporting and Data System (CO-RADS) score was used to detect COVID-19 infected patients more effectively than the RT-PCR test specifically when the patient reported COVID-19 symptoms but tested RT-PCR negative [34–36]. The present study attempted to correlate the high CO-RADS score recorded at the time of admission with the biochemical test parameters. Though a direct correlation with severity was not observed, 92.4% of ICU

admitted patients reported high CO-RADS score (≥4). A recent study investigated the prediction model of the chest CT score using CO-RADS to diagnose COVID-19 with better specificity and positive predictive value compared to RT-PCR test [37].

The ROC plot independently and in the combination of CO-RADS with biochemical features ferritin and blood urea nitrogen values revealed that cumulative use of the radiological and biochemical factors (AUC = 0.772) could better predict ICU admission among the COVID-19 positive UAE patients. This could be an ideal alternative to decide on immediate intensive care admission especially when the PCR test result is suspicious and a repeat test was advised after seven days interval. CO-RADS has been already suggested as a primary diagnostic tool where RT-PCR test result may take time and the patient is admitted with critical vitals [34, 38–40].

The single centered observational study presented here reports for the first time in the Gulf region, a cumulative parameter including CO-RADS, serum ferritin, and blood urea nitrogen as a predictor for ICU admission among COVID-19 infected patients. The other important outcome from the study is that patients admitted with a reported history of chronic kidney disease are also proposed to be at high risk of severe infections. Also, patients presenting symptoms related to respiratory failure ARDS and dyspnea must be immediately tested for elevated levels of neutrophils, ferritin, and blood urea nitrogen which may inform the critical condition.

The main limitation of the study is that it was carried out in a single center in Sharjah, UAE. However, the present study reflected most of the findings from a metanalysis performed on fifty-eight such retrospective studies reported between January 2020 to December 2020 [41]. This systematic review informed about the risk of mortality among COVID-19 infected patients. In general, large number of studies dealt with prediction models for mortality rather than factors distinguishing ICU admitted patients. The unique observation from the present study was to correlate radiological data with biochemical laboratory findings that could help the clinician to isolate the patients immediately to intensive care and clinically manage with ICU specific protocols such as oxygen treatment, mechanical ventilation, and high flow nasal cannula. There were no reports, to the best of our knowledge, that performed such correlation. Most importantly, these findings could clearly differentiate the ICU admitted cases from the other COVID-19 infected patients which can expedite the clinical decisions upon admission. Possibly, a prediction model developed based on a larger cohort may validate the current findings.

## 5. Conclusion

In conclusion, COVID-19 infected patients diagnosed with high CO-RADS score (CO-RADS 4 and 5) and elevated levels of ferritin (>290 μg/mL) and blood urea nitrogen (>6.7 mg/dL) need ICU admission. The accuracy of ICU prediction is high (94.2%) when all the three parameters were considered together than regarded independently. A history of chronic kidney disease and symptoms such as ARDS can be additional factors to predict ICU admission. The findings from the study are useful to develop a clinical management strategy for ICU admission in healthcare centers.

## Supporting information

**S1 Table. Pre-existing comorbidities among the COVID-19 positive patients demarcated into non-ICU and ICU admitted.**
(DOCX)

**S2 Table. Treatment or clinical management for COVID-19 positive patients demarcated into non-ICU and ICU admitted.**
(DOCX)

## Author Contributions

**Conceptualization:** Riyad Bendardaf, Salah Abusnana.

**Data curation:** Zainab Al-Abadla, Dima Zein, Noura Alkhayal, Ramy Refaat Georgy, Feda Al Ali, Alaa Elkhider.

**Formal analysis:** Poorna Manasa Bhamidimarri, Dima Zein, Ramy Refaat Georgy.

**Funding acquisition:** Riyad Bendardaf.

**Investigation:** Poorna Manasa Bhamidimarri, Dima Zein, Noura Alkhayal, Ramy Refaat Georgy, Alaa Elkhider, Sadeq Qadri.

**Methodology:** Poorna Manasa Bhamidimarri, Zainab Al-Abadla, Dima Zein, Noura Alkhayal, Ramy Refaat Georgy, Feda Al Ali, Alaa Elkhider, Sadeq Qadri.

**Project administration:** Riyad Bendardaf, Rifat Hamoudi, Salah Abusnana.

**Resources:** Zainab Al-Abadla, Noura Alkhayal, Feda Al Ali, Alaa Elkhider, Sadeq Qadri.

**Software:** Ramy Refaat Georgy, Feda Al Ali.

**Supervision:** Rifat Hamoudi, Salah Abusnana.

**Validation:** Poorna Manasa Bhamidimarri, Rifat Hamoudi.

**Visualization:** Ramy Refaat Georgy.

**Writing – original draft:** Poorna Manasa Bhamidimarri.

**Writing – review & editing:** Riyad Bendardaf, Poorna Manasa Bhamidimarri, Rifat Hamoudi, Salah Abusnana.

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
