## [Decision Letter · Decision Letter 0]

25 Nov 2021

PONE-D-21-26443Ferritin, blood urea nitrogen and high chest CT score determines ICU admission in COVID-19 positive UAE patients: a single center retrospective studyPLOS ONE

Dear Dr. Hamoudi,

Thank you for submitting your manuscript to PLOS ONE. After careful consideration, we feel that it has merit but does not fully meet PLOS ONE’s publication criteria as it currently stands. Therefore, we invite you to submit a revised version of the manuscript that addresses the points raised during the review process.

We look forward to receiving your revised manuscript.

Kind regards,

Alexandra Lucas

Academic Editor

PLOS ONE

Journal Requirements:

Additional Editor Comments:

One reviewer has rejected this manuscript based on prior larger studies. One reviewer has requested only minor revisions but raises some concerns.

With the note we are designating this manuscript as requiring major revisions and we are asking the authors to respond to all critiques in a point by point fashion. Please reply carefully and in detail to each of the reviewers' comments in a point by point manner.

Reviewers' comments:

Reviewer's Responses to Questions

**Comments to the Author**

1. Is the manuscript technically sound, and do the data support the conclusions?

Reviewer #1: Partly

Reviewer #2: Yes

2. Has the statistical analysis been performed appropriately and rigorously? 

Reviewer #1: No

Reviewer #2: Yes

3. Have the authors made all data underlying the findings in their manuscript fully available?

Reviewer #1: Yes

Reviewer #2: Yes

4. Is the manuscript presented in an intelligible fashion and written in standard English?

Reviewer #1: Yes

Reviewer #2: Yes

5. Review Comments to the Author

Reviewer #1: While this is an interesting study the sample size, especially of sub analysis are very small. As a single center study the generalized applicability of these observations is limited. Some of the sub study groups have only 2 subjects and several recent publications on the same have 500 plus subjects available for analysis.

Reviewer #2: This manuscript by Bendardaf and Bhamidimarri et al., proposed predictive metrics for evaluating whether patients will be admitted to the ICU based on biochemical and radiological findings. This is a retrospective analysis. These findings are of importance and global relevance and may be useful for informing medical decisions in some cases.

The main concern is the small sample size (N=57). To their credit, the authors acknowledge this limitation in the manuscript, but the broader implication for such a small sample size is the potential to over-train their analysis. Essentially, if all 57 patients were used in the creation of the ROC curves, how do we know whether the identified metrics will be any good in the real world? ROC analysis is best when there is a "discovery" step followed by a "validation" step. With such a small sample size, it appears the authors have only done the "discovery" step. Can the authors reach out to another medical center to evaluate whether their patient data can be tested to validate the metrics identified here? This information might also be available from mining in the literature. The risk here is that these metrics might only be relevant to these 57 patients at University Hospital Sharjah.

6. PLOS authors have the option to publish the peer review history of their article (what does this mean?). If published, this will include your full peer review and any attached files.

Reviewer #1: No

Reviewer #2: No

---

## [Author Response · Author response to Decision Letter 0]

27 Feb 2022

PONE-D-21-26443

Reviewer #1: While this is an interesting study the sample size, especially of sub analysis are very small. As a single center study the generalized applicability of these observations is limited. Some of the sub study groups have only 2 subjects and several recent publications on the same have 500 plus subjects available for analysis.

Answer: We thank the reviewer for the positive comments about the study and agree that the sample size might be small to arrive at solid conclusion. In order to address the sample size limitation, we included ~50 additional cases in the ICU admitted group from the single center. In total, the current study has 66 in ICU admitted group (earlier it was 17) and 40 in non-ICU admitted group. The outcome observed in the earlier small cohort were similar to larger cohort. Moreover, the accuracy for predicting the ICU admission increased to 94.2% with inclusion of additional 50 cases in ICU admitted group for cumulative radiological and biochemical parameters than the earlier cohort. The conclusions largely remained the same in both the cases (earlier and now) indicating CORADS along with ferritin and blood urea nitrogen values determines the ICU admission more accurately than considered independent.

 In most of the studies, more specifically in relation to COVID-19, clinical characterization from single center study was helpful in determining the factors associated with COVID-19 which was correlating with the other large cohort studies. For example the observations made in a small cohort of 60 COVID-19 patients in China (https://journals.lww.com/md-journal/Fulltext/2021/07300/A_retrospective_analysis_from_a_single_center_for.37.aspx) were parallel with the large cohort of 1296 patients from Brazil both being single center studies (https://journal.einstein.br/article/clinical-characteristics-and-outcomes-of-covid-19-patients-admitted-to-the-intensive-care-unit-during-the-first-year-of-the-pandemic-in-brazil-a-single-center-retrospective-cohort-study/) 

In addition, meta-analysis conducted on 58 independent studies on COVID-19 showed the similar findings for the biochemical parameters from the current study (https://www.ncbi.nlm.nih.gov/pmc/articles/PMC8444810/). 

Reviewer #2: This manuscript by Bendardaf and Bhamidimarri et al., proposed predictive metrics for evaluating whether patients will be admitted to the ICU based on biochemical and radiological findings. This is a retrospective analysis. These findings are of importance and global relevance and may be useful for informing medical decisions in some cases.

The main concern is the small sample size (N=57). To their credit, the authors acknowledge this limitation in the manuscript, but the broader implication for such a small sample size is the potential to over-train their analysis. Essentially, if all 57 patients were used in the creation of the ROC curves, how do we know whether the identified metrics will be any good in the real world? ROC analysis is best when there is a "discovery" step followed by a "validation" step. With such a small sample size, it appears the authors have only done the "discovery" step. 

Can the authors reach out to another medical center to evaluate whether their patient data can be tested to validate the metrics identified here? This information might also be available from mining in the literature. The risk here is that these metrics might only be relevant to these 57 patients at University Hospital Sharjah.

Answer: We thank the reviewer for taking time to thoroughly review the article and identify the importance of the study.

As mentioned, we do agree about the sample size and hence addressed it by collecting information from additional 50 cases from ICU admitted group. In total, the current study has 66 in ICU admitted group (earlier it was 17) and 40 in non-ICU admitted group. The biochemical characteristics and symptoms correlated with the earlier smaller cohort. The important findings regarding cumulative use of CORADS along with ferritin and blood urea nitrogen values to determine ICU admission holds true even upon increasing the samples size. In fact, the current cohort showed an increase in accuracy to 94.2% when all the three parameters were combined to determine ICU admission. This indicated that the observations made in 57 samples were also relevant in 106 samples and are in coherence with other such studies performed at different centers.

In order to assess the findings in other study, we looked for similar studies but information on all the variables were not provided to pair the samples with corresponding CORADS score and biochemical characteristics. Hence, we had to rely mostly on the conclusions from independent studies. However, most of the observations made in our study reflected in other COVID-19 related studies which were also discussed in the article to correlate the findings from similar studies in another cohort.

---

## [Editor Report · Decision Letter 1]

17 May 2022

Ferritin, blood urea nitrogen and high chest CT score determines ICU admission in COVID-19 positive UAE patients: a single center retrospective study

PONE-D-21-26443R1

Dear Dr. Hamoudi,

We’re pleased to inform you that your manuscript has been judged scientifically suitable for publication and will be formally accepted for publication once it meets all outstanding technical requirements.

Kind regards,

Alexandra Lucas

Academic Editor

PLOS ONE

Additional Editor Comments (optional):

We apologize for the delayed response, the reviewers' reviews were delayed. We had requested a re review from reviewer #2 but they had suggested minor revisions and have not responded. You have responded appropriately to the comments from the reviewers and we believe this manuscript is acceptable for publication. We thank you for your patience and we are pleased to accept your manuscript.

---

## [Editor Report · Acceptance letter]

10 Jul 2022

PONE-D-21-26443R1 

Ferritin, blood urea nitrogen, and high chest CT score determines ICU admission in COVID-19 positive UAE patients: a single center retrospective study 

Dear Dr. Hamoudi:

I'm pleased to inform you that your manuscript has been deemed suitable for publication in PLOS ONE. Congratulations! Your manuscript is now with our production department. 

Kind regards, 

on behalf of

Professor Alexandra Lucas 

Academic Editor

PLOS ONE